# Observation of nonlinear thermoelectric effect in MoGe/Y$_3$Fe$_5$O$_{12}$

Hiroki Arisawa [1,2,3], Yuto Fujimoto[1], Takashi Kikkawa [1] & Eiji Saitoh [1,2,4,5] ✉

Thermoelectric effects refer to the voltage generation from temperature gradients in condensed matter. Although various power generators are made from them, all the known effects, such as Seebeck effect, require macroscopic temperature gradients; since the sign of the generated voltage is reversed by reversing the temperature gradient, the net voltage disappears when the temperature distribution fluctuates temporarily or spatially with a macroscopic temperature gradient of zero. It is impossible to utilize such temperature fluctuations in the conventional thermoelectric effects, a situation which limits their application. Here we report the observation of a second-order nonlinear thermoelectric effect; we develop a method to measure nonlinear thermoelectricity and observe that a superconducting MoGe film on Y$_3$Fe$_5$O$_{12}$ generates a voltage proportional to the square of the applied temperature gradient. The nonlinear thermoelectric generation demonstrated here provides a way for making power generators that produce electric power from temperature fluctuations.

Typical thermoelectric effects established so far are Seebeck effect[1] and Nernst effect[2]. Seebeck effect refers to the voltage generation along the temperature gradient, while Nernst effect refers to the voltage generation normal to the gradient. In both, the amplitude of the generated voltage is proportional to that of the temperature gradient, and disappears in the absence of macroscopic temperature gradients of a scale larger than the sample size.

Here we report the observation of second-order nonlinear thermoelectric voltages, whose amplitude is proportional to the square of the temperature gradient. To explore the nonlinear thermoelectric effect, a superconductor molybdenum-germanium (MoGe) on a magnet is a useful materials system[3]. It is a typical system showing nonlinear resistance that can be controlled by using magnetic fields. In MoGe, a superconducting vortex-liquid state appears[4,5] (see the phase diagram in Fig. 1d, determined[6] based on the resistivity measurement), in which a vortex, a string-shaped defect of superconductivity, can easily move by applying a current. The vortex motion driven by a current induces temporal changes in the local phase of the superconducting wave function around the vortex core and gives rise to the electromotive force due to the phase change. Therefore, the vortex liquid phase exhibits finite resistance in spite of its superconductivity[7] (see Fig. 1c). In MoGe films on a ferrimagnetic insulator Y$_3$Fe$_5$O$_{12}$ (YIG), the in-plane vortex resistance is externally nonlinear with respect to the current when a magnetic field is applied in the in-plane direction[3]. The nonlinear resistance has been attributed to the asymmetry of magnetic fluctuations between the top and bottom surfaces of the MoGe film (Fig. 1b), which causes different vortex nucleation rates[8] (Bean-Livingston barriers) between the surfaces and gives rise to different resistance depending on the vortex flow directions. The field-induced transition to the vortex liquid phase is so sharp that this nonlinear resistance can be turned on and off by changing the fields, an advantage which realizes controllable experiments. By using this materials system, as shown in Fig. 1a, b, we investigate a nonlinear thermoelectric effect, where the thermoelectric voltage due to the vortex Nernst effect[9] becomes nonlinear with respect to the temperature difference.

## Results

### Sample and measurement setup
The key for sensitively measuring a thermoelectric effect is the use of higher-harmonic lock-in methods[10,11]. In conventional linear

[1]Department of Applied Physics, The University of Tokyo, Tokyo 113-8656, Japan. [2]RIKEN Center for Emergent Matter Science, Wako 351-0198, Japan. [3]Institute for Materials Research, Tohoku University, Sendai 980-8577, Japan. [4]Institute for AI and Beyond, The University of Tokyo, Tokyo 113-8656, Japan. [5]WPI, Advanced Institute for Materials Research, Tohoku University, Sendai 980-8577, Japan. ✉e-mail: eizi@ap.t.u-tokyo.ac.jp

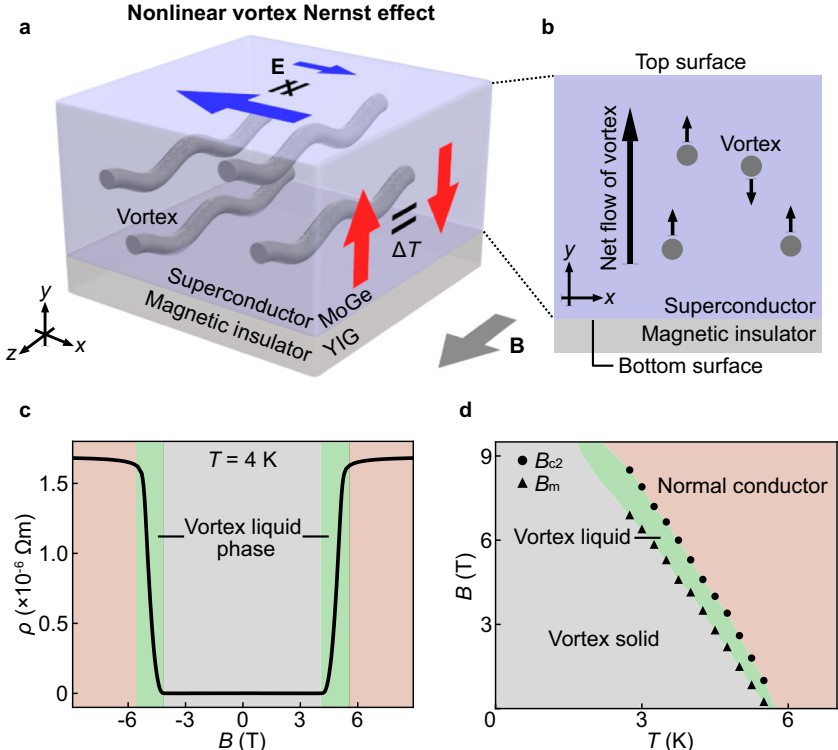

**Fig. 1 | Concept of nonlinear vortex Nernst effect. a** A schematic illustration of a nonlinear vortex Nernst effect. **E**, $\Delta T$, and **B** denote an electric field, a temperature difference, and an external magnetic field, respectively. The bottom surface of a superconductor is in contact with a magnetic insulator. **E** due to the vortex Nernst effect is nonlinear with respect to $\Delta T$ applied along the $y$ direction. The direction of **E** is perpendicular to the temperature gradient and the magnetic field. **b** The cross-sectional area (the $x$-$y$ plane) of the superconductor/magnetic-insulator bilayer system. Vortex strings are depicted by gray dots. **c** The $B$ dependence of the resistivity $\rho$ of a type-II superconductor MoGe film on a ferrimagnetic insulator

$Y_3Fe_5O_{12}$ (YIG) at $T = 4$ K. The green shaded area represents the magnetic field region of the vortex liquid phase, where vortex strings can move freely. The gray (red) shaded area shows the field region of the vortex solid (normal conductor) phase. **d** The $B$-$T$ phase diagram of the MoGe. The green, gray, and red shaded areas represent the vortex liquid, vortex solid, and normal conductor phases, respectively. $B_{c2}$ (black dots) and $B_m$ (black triangles) are the upper critical field and the vortex-solid melting field, respectively, determined by using the conditions[6] $\rho(B_{c2}) = 0.95\rho_N$ and $\rho(B_m) = 10^{-3}\rho_N$, where $\rho_N$ is the resistivity of the MoGe in the normal conducting state at $B = 9$ T.

thermoelectric measurement, an a.c. current, $I\sin\omega t$, is applied to a heater attached to an end of the sample while the other end is connected to a heat bath. Since the Joule heating is proportional to the square of the applied current, the temperature difference $\Delta T$ across the sample is proportional to $I^2\sin^2\omega t \propto \cos 2\omega t + \text{constant}$. Therefore, the linear thermoelectric voltage ($\propto \Delta T$) temporally oscillates at $2\omega$, which can be sensitively detected by a $2\omega$ lock-in method using the a.c. current as a reference.

Then, how can we measure the nonlinear thermoelectric voltage proportional to $(\Delta T)^2$ sensitively? One might think that a similar $4\omega$ lock-in method could be used, since the nonlinear thermoelectric voltage $[\propto (\Delta T)^2] \propto I_H^4$, where $I_H$ refers to the current applied to the heater and $\Delta T \propto I_H^2$. However, we found that the $4\omega$ lock-in method does not work properly, because the $4\omega$ harmonic component is contaminated by the multiplication of linear thermoelectric responses ($\propto I_H^2$); the order of the lock-in detection should be carefully chosen for reliable measurement (see Methods for details). Here we have developed a second-order lock-in method to measure the nonlinear thermoelectric voltages. First, two heaters are attached to both ends of the sample (Fig. 2d). Then, a.c. currents with the same frequency $\omega/2\pi$ are applied to the heaters, superimposed with d.c. currents. Owing to the d.c. currents, the heat generated at each heater contains a $1\omega$-oscillating component as an a.c.-d.c. cross term as well as a $2\omega$-oscillating component. The values of the heater currents are tuned so that the same amount of heat is generated from the two heaters (see Methods for details). When the phase difference $\phi$ between these heater currents is zero, an identical amount of heat is generated at the two heaters all the time, and the temperature difference across the sample

is kept zero. This condition can be confirmed by the disappearance of the thermoelectric voltage at $1\omega$ and $2\omega$. After establishing the condition, the relative phase $\phi$ between the heater currents is displaced from zero. When $\phi = \pi$, the two heaters are heated alternately at $1\omega$ (the $2\omega$-oscillating temperature difference remains zero since $2\phi = 2\pi$), and the temperature difference oscillating at $1\omega$ finally appears across the sample without $2\omega$ components. The second-order nonlinear thermoelectric effect converts this $1\omega$ temperature difference into a $2\omega$-oscillating voltage, which can be detected in terms of the $2\omega$ lock-in method by using the heater current as a reference. The expected $\phi$ dependence of this $2\omega$ lock-in voltage $V_{2\omega}$ is (see Fig. 2a, b, and Supplementary Note 1)

$$V_{2\omega} \propto \cos\phi\sin^2\left(\frac{\phi}{2}\right). \quad (1)$$

This characteristic dependence provides a touchtone for discriminating nonlinear thermoelectric effects.

Figure 2c shows a schematic illustration of the sample system used in the present study. The sample is an amorphous MoGe film with the thickness 150 nm, sputtered on YIG. YIG exhibits large magnetic susceptibility, resulting in a difference in the electromagnetic fluctuation amplitude between the top and bottom surfaces of the MoGe[3]. To apply $\Delta T$ along the $y$ axis, the sample was sandwiched by two heaters (Fig. 2d). Following the above procedure, we measured the second harmonic lock-in voltage $V_{2\omega}$ along the $x$ axis between the electrodes (Fig. 2c, d) using the current $I_{H1}\sin(\omega t + \phi)$ applied to the Heater-1 as a reference. We also applied the current $I_{H2}\sin\omega t$ to

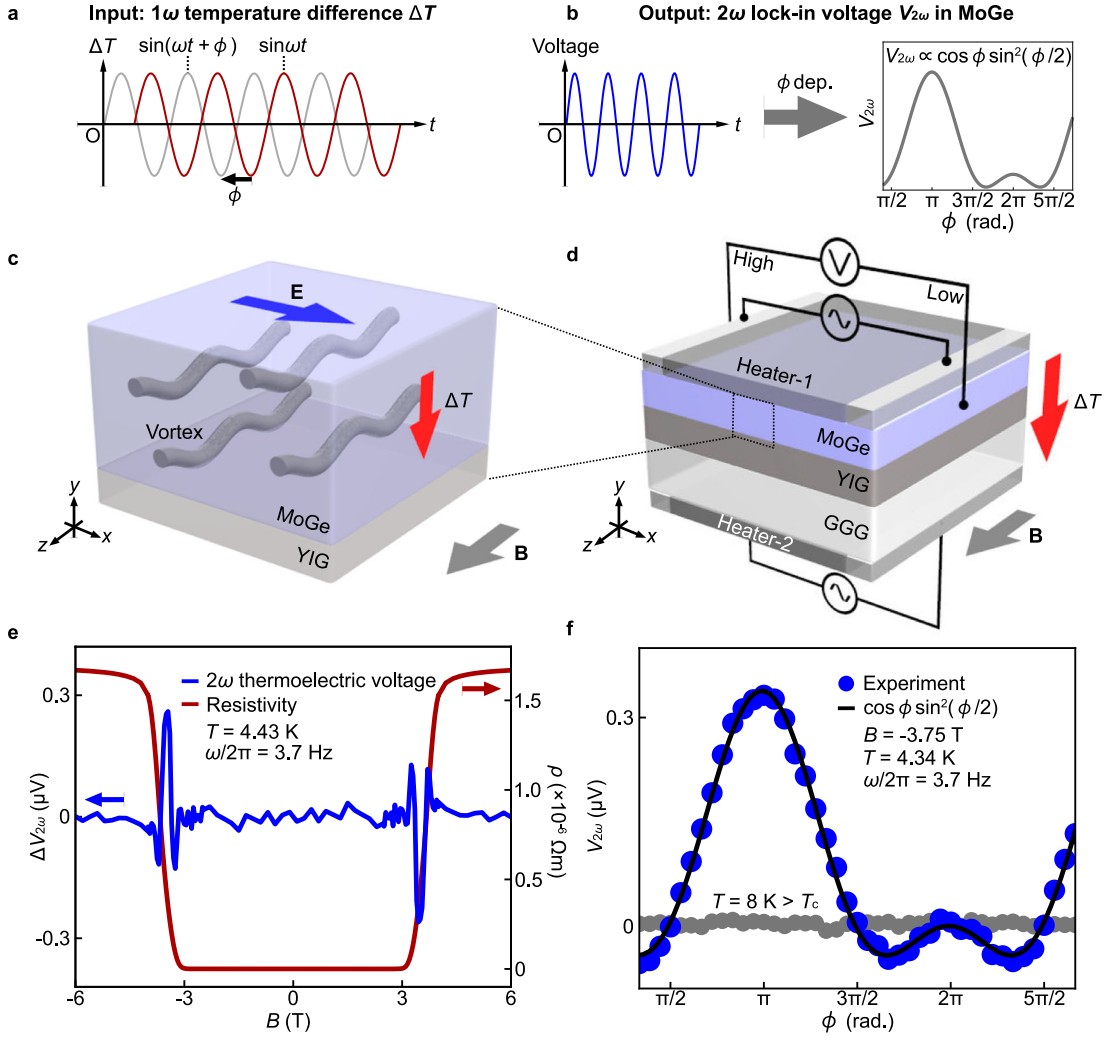

**Fig. 2 | Observation of nonlinear vortex Nernst effect in MoGe. a, b** The second-order lock-in method for detecting a nonlinear thermoelectric voltage. **a** An a.c. temperature difference $\Delta T$ between the top and bottom surfaces is applied to the MoGe with the frequency $\omega/2\pi$ as an input, and **b** the $2\omega$ lock-in voltage $V_{2\omega}$ synchronized with the a.c. $\Delta T$ is measured for the MoGe with changing the relative phase $\phi$ between the heater currents applied to the Heater-1 and the Heater-2. **c** A MoGe/YIG bilayer system used in the present study. In the vortex liquid phase of the MoGe, $\mathbf{E}$ due to the vortex Nernst effect appears in the MoGe perpendicular to the temperature gradient and the magnetic field ($x$ direction). **d** The measurement setup used in the present study. The MoGe/YIG bilayer film on a Gd$_3$Ga$_5$O$_{12}$ (GGG)

substrate was sandwiched by two heaters. The currents $I_{HI}\sin(\omega t + \phi) + I_{DC1}$ and $I_{H2}\sin\omega t + I_{DC2}$ were applied to the Heater-1 and the Heater-2, respectively. The phase difference $\phi$ between the heater currents was changed and $V_{2\omega}$ was measured for the MoGe along the $x$ axis in response to $\Delta T$ under **B**. **e** The $B$ dependence of $\Delta V_{2\omega} = V_{2\omega}(\phi) - V_{2\omega}(\phi + \pi)$ at $\phi = \pi$ (blue solid curve) and the resistivity $\rho$ (red solid curve) measured for the MoGe. The values of $T$ and $\omega/2\pi$ were set to 4.43 K and 3.7 Hz, respectively. **f** The $\phi$ dependence of $V_{2\omega}$ for the MoGe at $B = -3.75$ T and $\omega/2\pi = 3.7$ Hz. Blue dots (Gray dots) and the black solid curve are experimental data at $T = 4.34$ K ($T = 8$ K) and a $\cos\phi\sin^2(\phi/2)$ curve, respectively. $T_c$ denotes the transition temperature of the superconductivity for the MoGe.

the Heater-2. Here $\phi$ is the phase difference between the two heater currents. We define the normalized second harmonic voltage as $\Delta V_{2\omega} = V_{2\omega}(\phi) - V_{2\omega}(\phi + \pi)$.

## Observation of nonlinear thermoelectric effect

In Fig. 2e, we show the second harmonic voltage $\Delta V_{2\omega}$ measured for the YIG/MoGe at 4.43 K and $\phi = \pi$ (blue solid curve), a condition where the $1\omega$-oscillating temperature difference across the sample is maximized. Importantly, a clear $\Delta V_{2\omega}$ signal appears at 3 T $<|B|<$ 4 T. This field range coincides with the appearance condition for the superconducting vortex-liquid phase in the MoGe at the present temperature (see the resistivity $\rho$ in Fig. 2e), signaling that the second-harmonic nonlinear signal $\Delta V_{2\omega}$ appears only when vortices are mobile.

Figure 2f shows the $\phi$ dependence of $V_{2\omega}$ measured with the fixed magnetic field $B = -3.75$ T at $T = 4.34$ K. With the change in $\phi$, $V_{2\omega}$

exhibits complicated oscillation. The observed $\phi$ dependence is in perfect agreement with the characteristic behavior, predicted for the nonlinear thermoelectric effect as Eq. (1) (see Fig. 2b, f). By tuning the phase $\phi$ to $\phi = 2\pi$, at which the temperature difference generated by the heaters is absent, the value of $V_{2\omega}$ decreases to $\sim 0$, showing that the observed $V_{2\omega}$ signal is caused by the temperature difference generated in the sample, not by the simple heating of the sample. The absence of the $V_{2\omega}$ signal in a MoGe/SiO$_2$ sample also shows that the influence of the sample heating is negligibly small (see Supplementary Note 3 for details). All these are evidence that the observed $V_{2\omega}$ signal is the second-order nonlinear thermoelectric voltage caused by vortex motion. Note that the $V_{2\omega}$ picks up the second-order response with respect to the temperature difference (see Supplementary Note 4 for details). The $V_{2\omega}$ signal disappears (gray dots in Fig. 2f) when the system temperature is higher than the transition temperature of the superconductivity, $T_c$, supporting the interpretation. We also

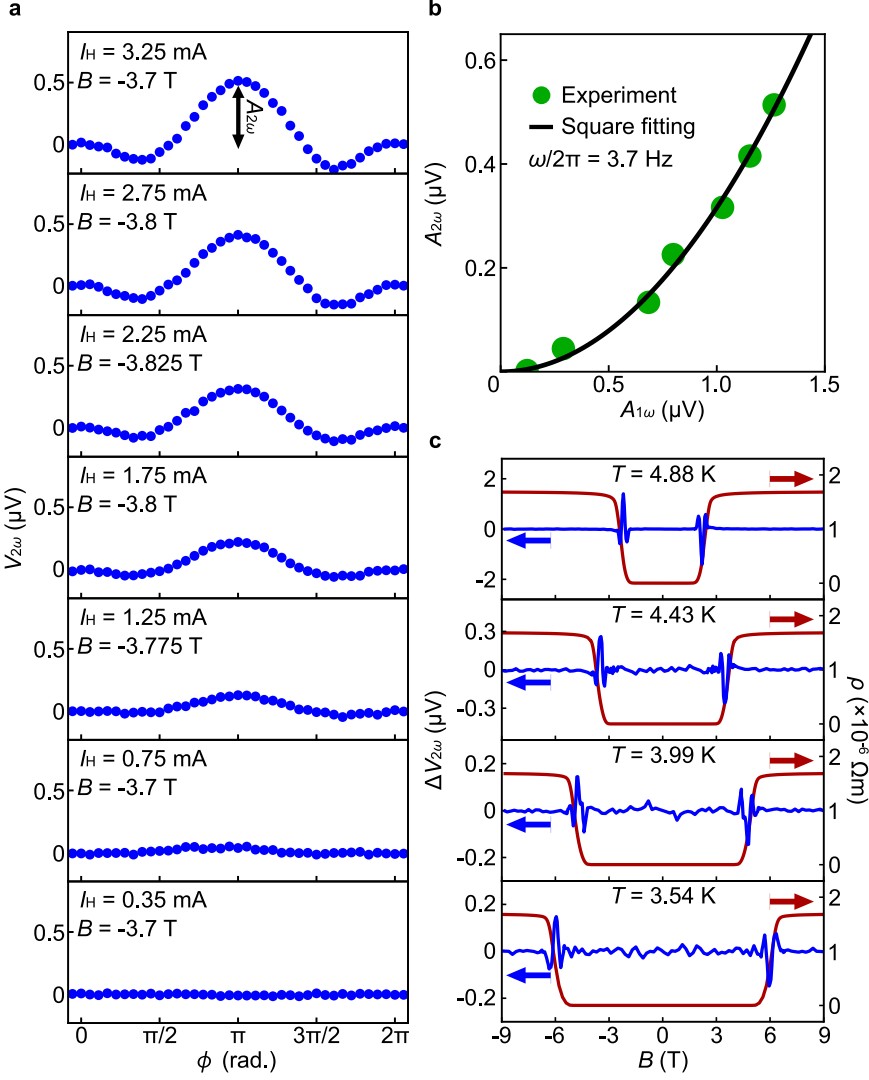

**Fig. 3 | Heater current and temperature dependence for nonlinear vortex Nernst effect. a** The $\phi$ dependence of $V_{2\omega}$ measured for the MoGe at each value of the a.c. heater-current amplitude $I_H = (I_{H1} + I_{H2})$, where the currents $I_{H1}\sin(\omega t + \phi) + I_{DC1}$ and $I_{H2}\sin\omega t + I_{DC2}$ were applied to the Heater-1 and the Heater-2, respectively. $A_{2\omega}$ is the peak amplitude of the $V_{2\omega}$ signal at $\phi = \pi$. **b** $A_{2\omega}$ as a function of the $1\omega$ lock-in voltage amplitude $A_{1\omega}$ (green dots). The black solid curve is a square fitting. **c** The $B$ dependence of $\Delta V_{2\omega}$ at $\phi = \pi$ (blue solid curves) and $\rho$ (red solid curves) at $T = 3.54$, $3.99$, $4.43$, and $4.88$ K. The value of $\omega/2\pi$ was set to 3.7 Hz in the measurement.

confirmed that the spatial temperature inhomogeneity is small along the sample length (see Supplementary Note 6 and 7 for details), and confirmed that the $V_{2\omega}$ signal appears at different values of $\omega/2\pi$ (see Supplementary Note 9 for details).

In Fig. 3a, we show the heater-current amplitude $I_H$ dependence of $V_{2\omega}$ for the YIG/MoGe. By increasing $I_H$ from 0.35 mA to 3.25 mA, the $V_{2\omega}$ signal proportional to $\cos\phi\sin^2(\phi/2)$ becomes more distinct with the peak amplitude $A_{2\omega}$. In Fig. 3b, we plot the $A_{1\omega}$ dependence of $A_{2\omega}$, where $A_{1\omega}$ refers to the peak amplitude of the $1\omega$ linear thermoelectric voltage, measured simultaneously by using the lock-in method. The linear thermoelectric voltage is proportional to the temperature gradient applied to the sample. The data in Fig. 3b shows that $A_{2\omega}$ is proportional to $A_{1\omega}^2$, demonstrating that the observed second harmonic voltage is proportional to the square of the applied temperature gradient.

Figure 3c shows the temperature $T$ dependence of $\Delta V_{2\omega}$ for the YIG/MoGe at $\phi = \pi$. With the increase in $T$ from 3.54 K, the $\Delta V_{2\omega}$ peak shifts to lower fields (blue solid curves in Fig. 3c). The field dependence of the resistivity $\rho$ of the sample at each $T$ is shown as the red solid curves in Fig. 3c, which shows that the $\Delta V_{2\omega}$ peaks appear concomitant

with the sudden change in $\rho$ signaling the appearance of the vortex liquid state at each temperature. The results support our interpretation that the second harmonic voltage is due to the vortex motion in the vortex liquid phase in the MoGe/YIG.

**Phenomenological model for nonlinear vortex Nernst effect**

We formulated the nonlinear vortex Nernst effect in the MoGe/YIG in terms of a phenomenological model[3]. In the model, we assume that the vortex flow $\mathbf{J}_v(+\Delta T)$ in the $-y$ direction [$\mathbf{J}_v(-\Delta T)$ in the $+y$ direction] due to the temperature difference $\Delta T$ [$-\Delta T$] is governed by the nucleation process[12,13] of vortex strings at the interfaces of the MoGe (Fig. 4a). The vortex nucleation rate $\propto e^{-\Delta F/k_B T}$ is determined by the nucleation energy barrier $\Delta F$ of a vortex at the interface[14], where $k_B$ is the Boltzmann constant. The energy barrier $\Delta F_{YIG}$ at the MoGe/YIG interface is less than that at the opposite interface, $\Delta F_{vac}$, since the greater magnetic susceptibility in YIG reduces the magnetostatic energy of the stray field created in the YIG by a nucleated vortex[3] (see Supplementary Note 2). The asymmetry of the energy barrier, $\Delta F_{YIG} < \Delta F_{vac}$, induces a difference in the vortex nucleation rate between the interfaces under $\Delta T$, and leads to an asymmetric thermal

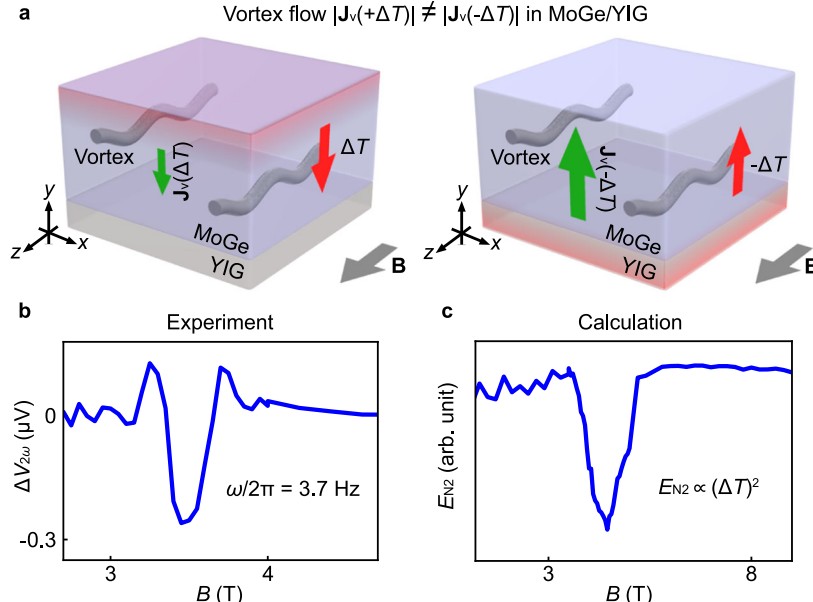

**Fig. 4 | Comparison between experiment and calculation. a** A schematic illustration of a theoretical model for the nonlinear vortex Nernst effect in the MoGe/YIG. A difference in the vortex nucleation rate between the top and bottom surfaces of the MoGe leads to the nonlinear vortex Nernst voltage. $\mathbf{J}_v(\Delta T)$ [$\mathbf{J}_v(-\Delta T)$] represents a vortex flow in the $-y$ [$+y$] direction induced by the temperature difference $\Delta T$ [$-\Delta T$]. **b** The $B$ dependence of $\Delta V_{2\omega}$ at $\phi = \pi$ measured for the MoGe/YIG. The values of $T$ and $\omega/2\pi$ were set to 4.43 K and 3.7 Hz, respectively. **c** The $B$ dependence of the electric field $E_{N2}$ due to the nonlinear vortex Nernst effect at $T = 4$ K calculated from Eq. (2) (see Supplementary Note 2 for details).

vortex flow $|\mathbf{J}_v(+\Delta T)| \neq |\mathbf{J}_v(-\Delta T)|$ (Fig. 4a). The asymmetric vortex flow generates an electric field according to $\mathbf{E} \propto \mathbf{B} \times \mathbf{J}_v$, giving rise to the vortex Nernst effect proportional to $(\Delta T)^2$: a nonlinear vortex Nernst effect. We calculated the electric field due to the nonlinear vortex Nernst effect as (see Methods and Supplementary Note 2 for details)

$$E_{N2} \propto \frac{S_\phi}{\eta d}\left(\frac{\Delta F_{YIG}}{k_B T} e^{-\frac{\Delta F_{YIG}}{k_B T}} - \frac{\Delta F_{vac}}{k_B T} e^{-\frac{\Delta F_{vac}}{k_B T}}\right)(\Delta T)^2, \quad (2)$$

where $S_\phi$, $\eta$, and $d$ are the entropy carried by a vortex, the vortex viscosity, and the thickness of the MoGe film, respectively. In Fig. 4c, we show the $B$ dependence of $E_{N2}$ calculated from Eq. (2), which reproduces the experimental data (see Fig. 4b). The measurement method presented here will allow the exploration of nonlinear thermoelectric effects in various candidate materials, such as bilayer conductors that exhibit nonreciprocal resistance[15] and polar ferromagnetic conductors[16].

The nonlinear thermoelectric effect can be used for making temperature-fluctuation sensors and voltage generators using out-of-equilibrium temperature fluctuations. The effect also bridges a gap between thermoelectricity and nonlinear physics in a solid. It will certainly be important to find matter that exhibits the effect at room temperature.

## Methods
### Sample preparation
A $Y_3Fe_5O_{12}$ (YIG) film with the thickness ~1 μm was grown on a $Gd_3Ga_5O_{12}$ (GGG) (111) substrate with the thickness 520 μm by liquid phase epitaxy. An amorphous MoGe film with the thickness 150 nm was sputtered on the YIG by RF sputtering in the Argon atmosphere of $2.8 \times 10^{-1}$ Pa. During the sputtering, the substrate was rotated at 3000 rpm and kept at room temperature by water cooling[17]. The length and the width of the MoGe/YIG sample are 8.0 mm and 1.0 mm, respectively.

### Measurement setup for nonlinear vortex Nernst effect
The measurement was performed by a standard lock-in technique[10,11] using Physical Property Measurement System (Quantum Design, Inc.). When a temperature difference, $\Delta T$, is applied to a material, the generated thermoelectric voltage $V$ can be expanded as $V = a_{1st}\Delta T + a_{2nd}(\Delta T)^2 + \cdots$, where $a_{n-th}$ ($n = 1, 2, \cdots$) is the $n$-th order thermoelectric coefficient. When the input temperature difference oscillates at the frequency $\omega/2\pi$, the $n$-th harmonic voltage detected with a lock-in amplifier is given by $V_{n\omega}(t) = \frac{1}{t_0}\int_{t-t_0}^{t}\sin(n\omega t' + \theta_0)V(t')dt'$, where $t_0$ is a sufficiently large averaging time and $\theta_0$ is a set phase for the amplifier.

In conventional linear thermoelectric measurement, a temperature difference is generated via the Joule heating induced by the a.c. current $\propto \sin \omega t$ applied to a heater attached to an end of the sample, $\Delta T \propto \sin^2\omega t$, and the $2\omega$ lock-in voltage is measured. Similarly, one might think that a nonlinear thermoelectric effect $\propto (\Delta T)^2$ can be measured via the $4\omega$ lock-in measurement. However, the $4\omega$ lock-in signal may include not only the second harmonic response but also the third harmonic one which may appear even without inversion symmetry breaking. To address this issue, the order of the lock-in detection should be lowered, which is achieved if the input temperature difference oscillates in a first-harmonic manner, $\Delta T \propto \sin \omega t$.

To create $\Delta T \propto \sin \omega t$ and detect the second-order nonlinear thermoelectric voltage selectively, two heaters (Heater-1 and Heater-2) were attached to the MoGe and GGG surfaces, respectively (Fig. 2d). The currents $I_{H1}\sin(\omega t + \phi) + I_{DC1}$ and $I_{H2}\sin \omega t + I_{DC2}$ were applied to the Heater-1 and the Heater-2, respectively. Due to the Joule heating, $\Delta T$ is generated in the MoGe film along the thickness direction ($y$ direction in Fig. 2c, d):

$$\Delta T = R_1\left[I_{H1}\sin(\omega t + \phi) + I_{DC1}\right]^2 - R_2\left(I_{H2}\sin \omega t + I_{DC2}\right)^2$$
$$= \tilde{I}_{DC1}^2 - \tilde{I}_{DC2}^2 + 2\tilde{I}_{DC1}\tilde{I}_{H1}\sin(\omega t + \phi) - 2\tilde{I}_{DC2}\tilde{I}_{H2}\sin \omega t \quad (3)$$
$$+ \tilde{I}_{H1}^2\sin^2(\omega t + \phi) - \tilde{I}_{H2}^2\sin^2\omega t.$$

Owing to the a.c. and d.c. currents, the resultant temperature difference contains the $1\omega$-oscillating components as the a.c.-d.c. cross terms. Here, $R_1$ ($R_2$) represents the prefactor relating the Joule heating of the Heater-1 (Heater-2) to $\Delta T$, and $\tilde{I}_{\text{H1,DC1}}$ ($\tilde{I}_{\text{H2,DC2}}$) is defined as $\tilde{I}^2_{\text{H1,DC1}} = R_1 I^2_{\text{H1,DC1}}$ ($\tilde{I}^2_{\text{H2,DC2}} = R_2 I^2_{\text{H2,DC2}}$). Before the measurement of the nonlinear vortex Nernst voltage, the values of $I_{\text{DC1}}$, $I_{\text{DC2}}$, $I_{\text{H1}}$, and $I_{\text{H2}}$ were tuned so that the same amount of heat is generated from the two heaters when $\phi = 0$, which is realized under $\tilde{I}_{\text{DC1}} = \tilde{I}_{\text{DC2}} = \tilde{I}_{\text{DC}}$ and $\tilde{I}_{\text{H1}} = \tilde{I}_{\text{H2}} = \tilde{I}_{\text{H}}$ (see Supplementary Note 1 for the detailed procedure). When the relative phase $\phi$ between the heater currents is changed from 0 to $\pi$, the two heaters are heated alternately at $1\omega$, and the temperature difference oscillating at $1\omega$ is produced across the sample without $2\omega$ components: $\Delta T \propto \sin \omega t$. The validity of this method is confirmed by measuring the generated temperature gradient with film-thermometers attached to the sample or by measuring the Nernst effect for a reference Permalloy sample (see Supplementary Notes 7 and 8 for details). The nonlinear vortex Nernst effect converts this $1\omega$ temperature gradient into a $2\omega$-oscillating transverse voltage $\propto (\nabla T)^2$, and the $2\omega$ lock-in voltage $V_{2\omega}^{\text{2nd}}$ should exhibit a characteristic $\phi$ dependence:

$$V_{2\omega}^{\text{2nd}} \propto a_{\text{2nd}} \cos \phi \sin^2\left(\frac{\phi}{2}\right), \tag{4}$$

which is different from the $2\omega$ "linear" vortex Nernst voltage $V_{2\omega}^{\text{1st}}$ arising from the fifth and sixth terms in Eq. (3):

$$V_{2\omega}^{\text{1st}} \propto a_{\text{1st}} \sin^2 \phi. \tag{5}$$

This shows that the nonlinear thermoelectric voltage can be distinguished from the conventional linear thermoelectric voltage in terms of the $\phi$ dependence of $V_{2\omega}$. In the experiments, the second harmonic lock-in voltage $V_{2\omega}$ in the MoGe in the $x$ direction is measured with a lock-in amplifier (LI5640, NF Corporation) by using the Heater-1 current $I_{\text{H1}} \sin(\omega t + \phi)$ as a reference. All the $\Delta V_{2\omega}$-$B$ and $V_{2\omega}$-$\phi$ data were anti-symmetrized with respect to the magnetic field $B$.

**Phenomenological model for nonlinear vortex Nernst effect**
We consider a superconducting vortex system in a MoGe film attached to YIG. In the vortex liquid phase, vortex strings can move freely in the MoGe under a driving force. For simplicity, we assume that the vortex flow is determined by the nucleation process of vortex strings at the MoGe surfaces[3]. The nucleation rate of a vortex is given by $P \propto e^{-\Delta F/k_{\text{B}}T}$, where $\Delta F$, $k_{\text{B}}$, and $T$ are the nucleation energy barrier of a vortex at the interface[14], the Boltzmann constant, and temperature. Since $\Delta F$ is affected by the magnetic susceptibility in the vicinity of the MoGe surfaces and YIG exhibits the larger susceptibility than that of vacuum, the nucleation energy barrier $\Delta F_{\text{YIG}}$ at the MoGe/YIG interface is less than that at the opposite MoGe/vacuum interface, $\Delta F_{\text{vac}}$, which results in the difference in $P$ between the top and bottom MoGe surfaces.

In the presence of the temperature difference $\Delta T$ (see Fig. 4a), vortex strings feel a thermal force, $\mathbf{f}_{\text{th}} = S_\phi \Delta T/d(-\mathbf{e}_y)$, where $S_\phi$, $d$, and ($-\mathbf{e}_y$) are the entropy carried by a vortex, the thickness of the MoGe film, and a unit vector along the $y$ axis, respectively. This thermal force accelerates a vortex string to a speed $\mathbf{v}$, and the vortex string feels a frictional force, $\mathbf{f}_{\text{f}} = -\eta \mathbf{v}$, where $\eta$ is the viscosity. In a steady state, the thermal force is in balance with the frictional force, $\mathbf{f}_{\text{th}} + \mathbf{f}_{\text{f}} = 0$, which leads to the terminal velocity of the vortex string $\mathbf{v}_{\text{v}} = S_\phi \Delta T/\eta d(-\mathbf{e}_y)$. When the temperature at the MoGe surface in contact with vacuum (YIG) is $T + \Delta T/2$ ($T - \Delta T/2$), the vortex flow $\mathbf{J}_{\text{v}}(\Delta T)$ in the $-y$ direction

becomes finite. The amplitude $J_{\text{v}}(\Delta T)$ of $\mathbf{J}_{\text{v}}(\Delta T)$ is described as

$$J_{\text{v}}(\Delta T) \propto \left[ P_{\text{vac}}\left( T + \frac{\Delta T}{2} \right) + P_{\text{YIG}}\left( T - \frac{\Delta T}{2} \right) \right] \frac{S_\phi}{\eta} \frac{\Delta T}{d}. \tag{6}$$

Here, $P_{\text{vac}}(T) \propto e^{-\Delta F_{\text{vac}}/k_{\text{B}}T}$ [$P_{\text{YIG}}(T) \propto e^{-\Delta F_{\text{YIG}}/k_{\text{B}}T}$] is the vortex nucleation rate at the MoGe/vacuum [MoGe/YIG] interface. The vortex flow generates an electric field according to $\mathbf{E} \propto \mathbf{B} \times \mathbf{J}_{\text{v}}$, and we calculate the electric field due to the nonlinear vortex Nernst effect $\propto (\Delta T)^2$ as

$$E_{\text{N2}} \propto \frac{(\Delta T)^2}{2} \frac{\partial^2}{\partial \Delta T^2} J_{\text{v}}(\Delta T) \bigg|_{\Delta T \to 0}. \tag{7}$$

By substituting Eq. (6) into Eq. (7), $E_{\text{N2}}$ is expressed as (see Supplementary Note 2 for details)

$$E_{\text{N2}} \propto \frac{S_\phi}{\eta d} \left( \frac{\Delta F_{\text{YIG}}}{k_{\text{B}}T} e^{-\frac{\Delta F_{\text{YIG}}}{k_{\text{B}}T}} - \frac{\Delta F_{\text{vac}}}{k_{\text{B}}T} e^{-\frac{\Delta F_{\text{vac}}}{k_{\text{B}}T}} \right) (\Delta T)^2. \tag{8}$$

## Data availability
The data that support the findings of this study are available from the corresponding author upon reasonable request.

## Code availability
The codes used in theoretical calculations are available from the corresponding author upon reasonable request.

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

## Acknowledgements

This work was supported by ERATO "Spin Quantum Rectification Project" (No. JPMJER1402) from JST, Japan, CREST (Nos. JPMJCR20C1 and JPMJCR20T2) from JST, Japan, Grant-in-Aid for Scientific Research (S) (No. JP19H05600) from JSPS KAKENHI, Japan, Grant-in-Aid for Scientific Research (B) (No. JP20H02599) from JSPS KAKENHI, Japan, Grant-in-Aid for Transformative Research Areas (No. JP22H05114) from JSPS KAKENHI, Japan, Grant-in-Aid for JSPS Fellows (No. JP20J21622) from JSPS KAKENHI, Japan, Murata Science Foundation, and Sumitomo Chemical. H.A. was supported by GP-Spin at Tohoku University.

## Author contributions

H.A. and Y.F. contributed equally to this work. H.A. and Y.F. prepared the samples and carried out the experiments with help from T.K. H.A. analyzed the data with help from Y.F. and T.K. H.A. and Y.F. formulated the theoretical model. E.S. planned and supervised the research. H.A., Y.F., and E.S. prepared the manuscript. All the authors discussed the results and commented on the manuscript.

## Competing interests

The authors declare no competing interests.
