## [Peer Review File · Nature Communications]

Observation of nonlinear thermoelectric effect in MoGe/
Y₃Fe₅O₁₂Editorial Note: This manuscript has been previously reviewed at another journal that is not operating a transparent peer review scheme. This document only contains reviewer comments and rebuttal letters for versions considered at *Nature Communications*.

REVIEWER COMMENTS

Reviewer #1 (Remarks to the Author):

It was great reviewing the manuscript again. The authors' changes address most of my questions and concerns. The Seebeck effect calibration helps exclude the issue of the Seebeck mixing, and I appreciate the adjustment of the significance statement. However, I do not follow the authors' response to the frequency dependence. All the original and the updated experimental results were carried out at 3.7 Hz. Have they quite tried at 1.7 Hz or at 37 Hz? If the interpretation is correct, shifting this frequency should not change the major results of the manuscript. And it would be a pretty straightforward test.

Given this concern, I suggest asking for a revision of the manuscript.

Reply to the Reviewer #1

We really thank the reviewer for the valuable comment on our manuscript. Following the comment, we carefully revised the manuscripts as shown below.

(Reviewer's comment)

It was great reviewing the manuscript again. The authors' changes address most of my questions and concerns. The Seebeck effect calibration helps exclude the issue of the Seebeck mixing, and I appreciate the adjustment of the significance statement. However, I do not follow the authors' response to the frequency dependence. All the original and the updated experimental results were carried out at 3.7 Hz. Have they quite tried at 1.7 Hz or at 37 Hz? If the interpretation is correct, shifting this frequency should not change the major results of the manuscript.

(Authors' response)

We thank the reviewer for the comment. Following the reviewer's instruction, we measured the nonlinear thermoelectric voltage at $\omega/2\pi = 1.7$ Hz and $\omega/2\pi = 37$ Hz.

Figure R1a shows the 2ω lock-in voltage $V_{2\omega}$ as a function of the phase ϕ of the 1ω -oscillating temperature difference ΔT for the MoGe/YIG at $\omega = 1.7$ Hz. $V_{2\omega}$ exhibits the same ϕ dependence [$\propto \cos\phi \sin^2(\phi/2)$] as $V_{2\omega}$ at $\omega/2\pi = 3.7$ Hz (Fig. R1b), which indicates the nonlinear thermoelectric effect $\propto (\Delta T)^2$. In contrast, $\omega/2\pi = 37$ Hz was actually too fast for the signal to be measured, because the time constant τ ($\gtrsim 40$ ms) of the thermal response of the sample is greater than the oscillation period of the a.c. heater current $2\pi/\omega = 1/(37 \text{ Hz}) \sim 27$ ms. Here, the lower limit of $\tau = CR_{\text{th}}$ ($\gtrsim 40$ ms) is roughly estimated from the heat capacity C and the thermal resistance R_{th} of the present sample system, where we assumed that the dominant contribution in C and R_{th} is the heat capacity of the GGG substrate [Schiffer P. *et al.*, Phys. Rev. Lett. **73**, 2500 (1994)] and the thermal resistance (in the order of 10 K/W) between the heater and the sample surface, respectively. When the value of $\omega/2\pi$ approaches $1/\tau$, the a.c. temperature difference in the sample cannot temporally follow the a.c. heater current and the sensitive lock-in detection of thermoelectric voltages becomes difficult.

Instead of $\omega/2\pi = 37$ Hz, we measured $V_{2\omega}$ at $\omega/2\pi = 13.7$ Hz, slightly less than $1/\tau$, and we confirmed that the $V_{2\omega}$ signal [$\propto \cos\phi \sin^2(\phi/2)$] indeed appears up to $\omega/2\pi = 13.7$ Hz (Figs. R1c-R1g). To make

this point clear, following the reviewer's comment, we added a sentence to the main text (Lines 97-99, Page 4) and a section to the revised Supplementary Information (Lines 385-395, Page 23).

Fig. R1 Frequency dependence of nonlinear thermoelectric voltage in MoGe/YIG. **a-g** The ϕ dependence of $V_{2\omega}$ at $\omega/2\pi =$ **(a)** 1.7 Hz, **(b)** 3.7 Hz, **(c)** 5.7 Hz, **(d)** 7.7 Hz, **(e)** 9.7 Hz, **(f)** 11.7 Hz, and **(g)** 13.7 Hz. The value of T was set to 4 K.

[Critical changes are listed below.]

- 1) The main text and Supplementary Information were revised in response to the reviewer's comment.
- 2) Section headings and subheadings were added to follow the format of the journal.
- 3) The abstract was changed and shortened to follow the format of the journal.
- 4) The final paragraph in Introduction was changed to follow the format of the journal.
- 5) The revised parts are highlighted in red color in the manuscript.
- 6) The order of Method, Date availability, Code availability, References, Acknowledgments, Author contributions, and Competing interests was rearranged to follow the format of the journal.

[Summary of revised parts in manuscripts in response to the reviewer's comment.]

Main text

Original (Lines 95-96, Page 4):

We also confirmed that the spatial temperature inhomogeneity is small along the sample length (see Supplementary Notes 6 and 7 for details).

Revised into (Lines 97-99, Page 4):

We also confirmed that the spatial temperature inhomogeneity is small along the sample length (see Supplementary Notes 6 and 7 for details), and confirmed that the $V_{2\omega}$ signal appears at different values of $\omega/2\pi$ (see Supplementary Note 9 for details).

Supplementary Information

Supplementary Note 9 was added (Lines 385-395, Page 23):

Supplementary Note 9 | Frequency dependence of thermoelectric voltage in MoGe/YIG sample.

Supplementary Figs. 14a-14g show the ϕ dependence of $V_{2\omega}$ for the MoGe/YIG at $\omega/2\pi = 1.7$ Hz, 3.7 Hz, 5.7 Hz, 7.7 Hz, 9.7 Hz, 11.7 Hz, and 13.7 Hz, respectively. $V_{2\omega}$ exhibits the characteristic ϕ dependence [$\propto \cos\phi \sin^2(\phi/2)$] for all values of ω , which indicates that the nonlinear thermoelectric effect $\propto (\Delta T)^2$ appears at the ω range.

Supplementary Fig. 14 Frequency dependence of nonlinear thermoelectric voltage in MoGe/YIG.

a-g The ϕ dependence of $V_{2\omega}$ at $\omega/2\pi =$ (a) 1.7 Hz, (b) 3.7 Hz, (c) 5.7 Hz, (d) 7.7 Hz, (e) 9.7 Hz, (f) 11.7 Hz, and (g) 13.7 Hz. The value of T was set to 4 K, and the value of B was set to -4.175 T (-4.225 T) in the measurement at $\omega/2\pi = 3.7$ Hz, 5.7 Hz, 7.7 Hz, 9.7 Hz, 11.7 Hz, and 13.7 Hz (the measurement at $\omega/2\pi = 1.7$ Hz).

REVIEWERS' COMMENTS

Reviewer #1 (Remarks to the Author):

The new revision addressed all my questions. I am glad my reviews helped in the revision. I recommend the acceptance of the manuscript.

Reply to the Reviewer #1

(Reviewer's comment)

The new revision addressed all my questions. I am glad my reviews helped in the revision. I recommend the acceptance of the manuscript.

(Authors' response)

We really thank the reviewer for the valuable comments on our manuscript and the recommendation to publish it in Nature Communications.